# Rapid recycling of glutamate transporters on the astroglial surface

**Piotr Michaluk[1,2]\*, Janosch Peter Heller[1,3], Dmitri A Rusakov[1]\***

[1]UCL Queen Square Institute of Neurology, University College London, London, United Kingdom; [2]BRAINCITY, Laboratory of Neurobiology, Nencki Institute of Experimental Biology PAS, Warsaw, Poland; [3]School of Biotechnology and National Institute for Cellular Biotechnology (NICB), Dublin City University, Glasnevin, Ireland

**Abstract** Glutamate uptake by astroglial transporters confines excitatory transmission to the synaptic cleft. The efficiency of this mechanism depends on the transporter dynamics in the astrocyte membrane, which remains poorly understood. Here, we visualise the main glial glutamate transporter GLT1 by generating its pH-sensitive fluorescent analogue, GLT1-SEP. Fluorescence recovery after photobleaching-based imaging shows that 70–75% of GLT1-SEP dwell on the surface of rat brain astroglia, recycling with a lifetime of ~22 s. Genetic deletion of the C-terminus accelerates GLT1-SEP membrane turnover while disrupting its surface pattern, as revealed by single-molecule localisation microscopy. Excitatory activity boosts surface mobility of GLT1-SEP, involving its C-terminus, metabotropic glutamate receptors, intracellular $Ca^{2+}$, and calcineurin-phosphatase activity, but not the broad-range kinase activity. The results suggest that membrane turnover, rather than lateral diffusion, is the main 'redeployment' route for the immobile fraction (20–30%) of surface-expressed GLT1. This finding reveals an important mechanism helping to control extrasynaptic escape of glutamate.

## Introduction

Excitatory transmission in the brain occurs mainly through the release of glutamate at chemical synapses. Once released, glutamate is taken up by high-affinity transporters that densely populate the plasma membrane of brain astrocytes (*Wadiche et al., 1995a*; *Danbolt, 2001*). The main glial glutamate transporter GLT1 (EAAT2) maintains extracellular glutamate at nanomolar levels, thus constraining its excitatory action mainly to the synaptic cleft (*Moussawi et al., 2011*; *Zheng and Rusakov, 2015*). Because synaptic vesicles release ~3000 glutamate molecules (*Savtchenko et al., 2013*) and because glutamate uptake cycle can take tens of milliseconds (*Wadiche et al., 1995b*), large numbers of transporter molecules have to be available near synapses to buffer the escaping glutamate (*Lehre and Danbolt, 1998*; *Bergles et al., 2002*). Indeed, the high occurrence of GLT1 in astroglial plasma membranes (*Danbolt, 2001*) ensures that regular network activity does not overwhelm glutamate transport (*Bergles and Jahr, 1998*; *Diamond and Jahr, 2000*). However, intense excitation can prompt glutamate escape from the immediate synapse, leading to activation of extrasynaptic receptors or even neighbouring synapses (*Lozovaya et al., 1999*; *Arnth-Jensen et al., 2002*; *Scimemi et al., 2004*; *Henneberger et al., 2020*; *Kopach et al., 2020*). Ultimately, the reduced availability of GLT1 in the synaptic environment has long been associated with pathologic conditions such as neurodegenerative diseases, stroke, or addiction (*Maragakis and Rothstein, 2004*; *Fontana, 2015*; *Kruyer et al., 2019*).

These considerations prompted intense interest in the cellular mechanisms underlying cellular trafficking and turnover of astroglial and neuronal glutamate transporters. A growing body of evidence has suggested the involvement of its carboxyl-terminal domain and protein kinase C (*Kalandadze et al., 2002*; *González et al., 2007*) and calmodulin-dependent protein kinase

\*For correspondence:
p.michaluk@nencki.edu.pl (PM);
d.rusakov@ucl.ac.uk (DAR)

**Competing interests:** The authors declare that no competing interests exist.

(*Underhill et al., 2015*), also engaging ubiquitin-dependent processes (*González et al., 2007*; *González-González et al., 2008*; *Martínez-Villarreal et al., 2012*) and constitutive protein sumoylation (*García-Tardón et al., 2012*; *Foran et al., 2014*; *Piniella et al., 2018*). Ultimately, these findings unveil the potential to regulate long-term, systemic changes in the GLT1 expression in a therapeutic context (reviewed in *Fontana, 2015*; *Peterson and Binder, 2019*). However, what happens to GLT1 trafficking on the time scale of the ongoing brain activity remains poorly understood. In recent elegant studies, single-particle tracking with quantum dots (QDs) has detected high surface mobility of GLT1 in astroglia (*Murphy-Royal et al., 2015*; *Al Awabdh et al., 2016*). Lateral diffusivity of transporters was boosted by local glutamatergic activity, thus suggesting the use-dependent surface supply of GLT1 towards active synapses (*Murphy-Royal et al., 2015*; *Al Awabdh et al., 2016*). However, synthetic QDs almost certainly prevent their link-labelled molecules from the membrane-intracellular compartment turnover and, at the same time, do not label any newly appearing molecules on the cell surface. Thus, the molecule-tracking observations relying solely on QDs could miss important changes in the composition and/or mobility of the studied molecular species due to their continuous recycling in the membrane.

We therefore set out to develop an approach enabling us to document, in real time, the exchange between membrane and intracellular fractions of GLT1, in addition to monitoring its lateral diffusion on the cell surface. To achieve this, we generated a fully functional variant of GLT1, termed GLT1-SEP, by adding an extracellular fragment with the pH-sensitive, Super-Ecliptic pHluorin (SEP); GLT1-SEP fluoresces when exposed to the extracellular medium but not in low pH of intracellular compartments. Expressing GLT1-SEP in astroglia in cell cultures and brain slices allowed us to combine the optical protocols of fluorescence recovery after photobleaching (FRAP) with molecular and pharmacological dissection, to monitor membrane turnover and lateral diffusion of the transporter proteins.

## Results

### Developing and probing GLT1-SEP

First, we designed the GLT1-SEP probe for FRAP measurements by introducing SEP into the second intracellular loop of GLT1a, between two proline residues (P199 and P200). GLT1a was selected because it accounts for 90% of total GLT1 expression in astrocytes in the rat brain, with GLT1b and GLT1c accounting for ~6% and ~1%, respectively (*Holmseth et al., 2009*). Next, aiming at astrocyte-specific expression, we cloned the construct under the gfaABC$_1$D promoter (*Lee et al., 2008*; *Figure 1A*; Materials and methods).

To test if this new construct (termed GLT1-SEP thereafter) is a functional glutamate transporter we transfected with it HEK 293T cells, which do not normally express GLT1. The control group of cells was transfected with the plasmid coding wild-type GLT1. For identification purposes, and to keep the same plasmid concentrations, cells were co-transfected with GLT1 constructs and mRFP1 under β-actin promoter, at a 2:1 ratio. Next, in whole-cell mode we recorded uptake currents in transfected cells induced by a 1 s application of 1 mM glutamate through a theta-glass solution-exchange system (*Figure 1B*), the method that avoids any mechanical concomitants of the application protocol (*Sylantyev and Rusakov, 2013*). Systematic recording across holding voltages produced normalised I-V curves that showed an excellent match between the wild-type native transporter and the mutant (*Figure 1C*). However, the absolute current in GLT1-SEP expressing cells was on average ~50% lower (*Figure 1—figure supplement 1A*), possibly because of the lower expression compared to native GLT1.

To understand further the relationship between the expression levels of native GLT1 and GLT1-SEP in HEK cells, we firstly compared immunostaining in cells expressing GLT1 or GLT1-SEP, and found that GLT1-SEP staining was indeed significantly lower (*Figure 1—figure supplement 2A,B*). Secondly, we compared expression of GLT1 variants in HEK cells using quantitative western blot (normalised against α-tubulin; n = 6 samples per condition). In these samples, GLT1 expression was twice as high as that of GLT1-SEP, and genetic deletion of the intracellular C-terminus (GLT1ΔC-SEP variant) did not change this difference significantly (*Figure 1—figure supplement 2C,D*). Thus, the decrease in the surface expression of GLT1-SEP and in its transporter current in HEK cells (compared

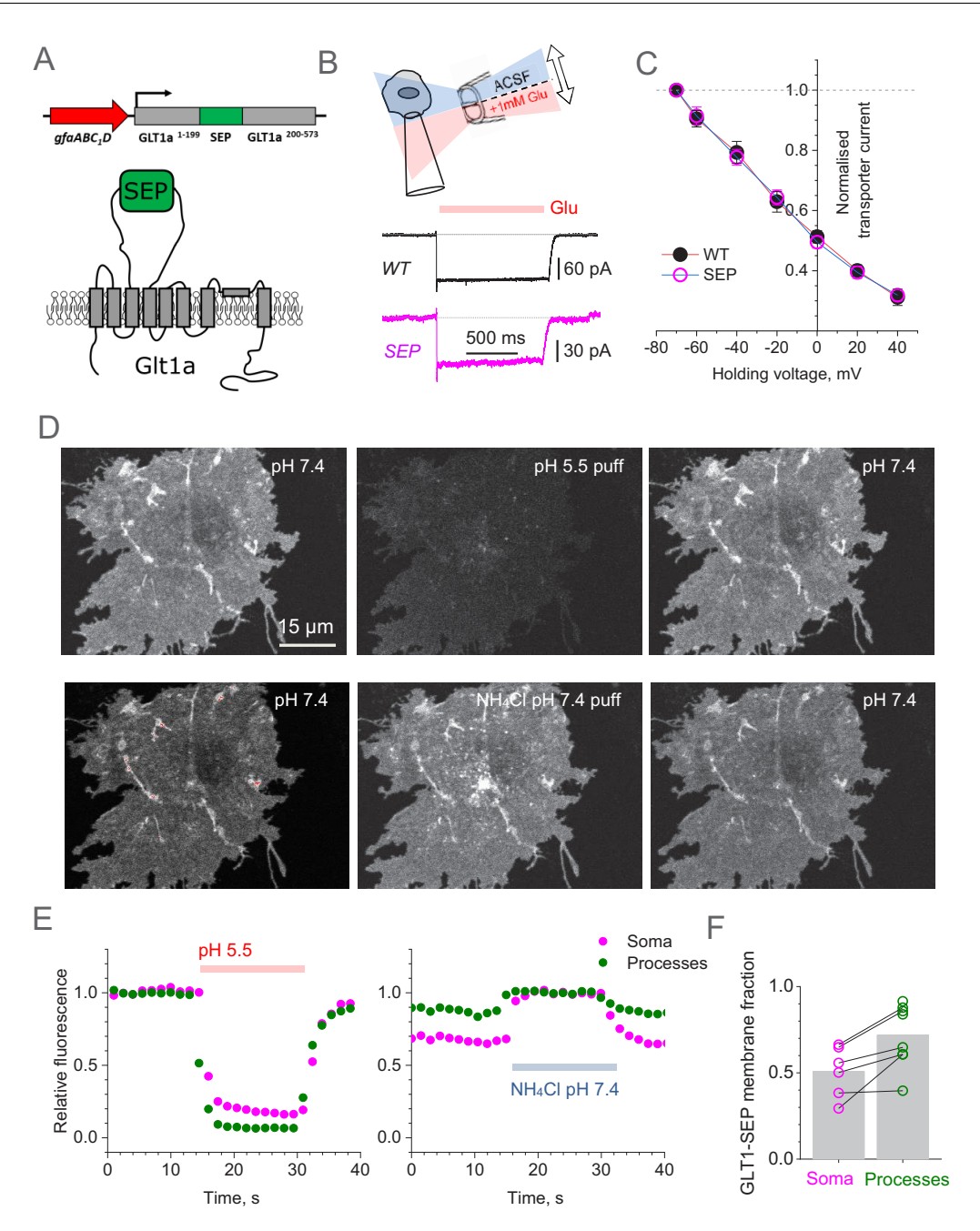

**Figure 1.** Super-Ecliptic pHluorin-tagged GLT1 (GLT1-SEP) enables monitoring of cell membrane and cytosolic fractions of glial glutamate transporters. (A) Diagram illustrating molecular composition of GLT1-SEP. (B) Functional probing of GLT1-SEP expressed in HEK cells shows a prominent current response to glutamate application, similar to that wild-type GLT1 (WT); top diagram, theta-glass pressure pipette application; traces, one-cell examples ($V_h = -70$ mV). (C) Summary of tests shown in (A): normalised current-voltage dependencies of GLT1 (mean ± SEM; n = 8) and GLT1-SEP (n = 4) are indistinguishable; current values normalised at $V_h = -70$ mV (absolute values 194 ± 29 pA and 81 ± 10 pA for GLT1 and GLT1-SEP, respectively). (D) Transient acidification (~10 s pH 5.5 puff, upper row) supresses cell-surface GLT1-SEP fluorescence whereas transient membrane $NH_4^+$ permeation (~10 s $NH_4Cl$ puff, lower row) reveals the cytosolic fraction of GLT1-SEP; one-cell example. (E) Time course of fluorescence intensity averaged over the cell soma (magenta) or all processes (green) in the test shown in (C). (F) Average cell-surface fraction R of GLT1-SEP (summary of experiments shown in D and E); dots, individual cells (connecting lines indicate the same cell); grey bars, average values (R mean ± SEM: 0.51 ± 0.15, n = 6 for somata; 0.72 ± 0.18, n = 8 for processes; soma boundaries in two cells were poorly defined).

The online version of this article includes the following source data and figure supplement(s) for figure 1:

**Source data 1.** Original data readout for *Figure 1D–F*, *Figure 1—figure supplement 2B,D* and *Figure 1—figure supplement 3B,C*.

**Figure supplement 1.** Testing glutamate transport function and the membrane/cytosol fraction ratio for GLT1-SEP.
**Figure supplement 2.** Expression of GLT1, GLT1-SEP, and GLT1ΔC-SEP in HEK cells.
**Figure supplement 3.** Relative expression GLT1, GLT1-SEP, and GLT1ΔC-SEP astrocytes.

with the case of wild-type GLT1) were matched, suggesting that adding the extracellular SEP tag had no detectable effect on glutamate uptake properties of GLT1.

## Intracellular versus membrane fractions of GLT1-SEP in astroglia

We next expressed GLT1-SEP in mixed cultures of neurons and glial cells. Thanks to the gfaABC$_1$D promoter, the probe was almost exclusively expressed in astrocytes. The living GLT1-SEP expressing cells were readily visualised, featuring a dense and homogenous expression pattern that reveals fine details of cell morphology (*Figure 1D*, upper left). Because the pH-sensitive GLT1-SEP fluoresces at higher extracellular pH but not at lower pH in intracellular compartments, we were able to estimate directly its membrane and intracellular fractions. Firstly, we confirmed that the observed fluorescence comes mainly from the membrane fraction of GLT-SEP. Indeed, brief acidification of the extracellular medium to pH 5.5 (10 s pipette puff with pH-adjusted bath medium) reversibly suppressed GLT1-SEP fluorescence (*Figure 1D*, upper row). Conversely, proton permeation of the cell membrane (10 s puff with NH$_4$Cl) could reveal both intra- and extracellular GLT1-SEP fractions, in a reversible fashion (*Figure 1D*, lower row). Systematic quantification of these experiments (*Figure 1E*) provided an estimate of the average GLT1-SEP surface fraction in astroglial processes, $R = 0.72 \pm 0.18$ (n = 8 cells, *Figure 1F*). In other words, between 2/3 and 3/4 of all cellular GLT1-SEP were exposed to the extracellular space. The $R$ estimate for the cell soma was somewhat lower (*Figure 1F*), but because exact identification of the somatic boundaries was ambiguous, we did not use somatic data in further analyses.

These data provided an important constraint for a (steady-state) quantitative assessment of the GLT1-SEP turnover kinetics. Introducing the membrane-intracellular exchange reaction (*Figure 1— figure supplement 1B*) as $C_{in} \overset{k_1}{\underset{k_{-1}}{\rightleftharpoons}} C_{mem}$ ($C_m$ and $C_{in}$ are membrane and intracellular concentration of GLT1-SEP, respectively) leads to a direct relationship between the corresponding kinetic constants $k_1$ and $k_{-1}$ (*Figure 1—figure supplement 1C*): $k_{-1} = \left(\frac{1}{R} - 1\right)k_1 = 0.389k_1$.

Again, to understand better how transfected and non-transfected astroglia express GLT1 variants, we stained transfected astrocytes with GFP in the green channel while using an antibody against GLT1 C-terminus in the red channel. We were thus able to compare all-variant GLT1 expression in the astrocytes transfected with GLT1-SEP (n = 29) with GLT1 expression in the neighbouring, non-transfected cells (*Figure 1—figure supplement 3A*). In the transfected astrocytes, the all-variant GLT1 level was higher than in non-transfected cells (*Figure 1—figure supplement 3B*). In the astrocytes transfected with the GLT1-SEP variant with the truncated cytosolic carboxy-terminal (GLT1ΔC-SEP, Materials and methods)) we were able to detect only endogenous GLT1 levels as the antibody does not recognise GLT1ΔC-SEP. In the transfected astrocytes, GLT1 level was 25–50% higher than that for GLT1 in non-transfected cells (*Figure 1—figure supplement 3B*). We also confirmed that the endogenous GLT1 expression was perfectly stable in non-transfected cells that are present in transfected culture samples (*Figure 1—figure supplement 3C*). These immunostaining data have simply confirmed that the membranes of both transfected and non-transfected cells are densely and relatively uniformly packed with the respective GLT1 variants.

Although the steady-state data obtained here are key for understanding the relationship between intracellular and membrane fractions of GLT1, they could not on their own reveal the actual rate of GLT1-SEP turnover in the cell membrane. To address this, we implemented a different approach, as described below.

## GLT1-SEP recycling in the plasma membrane

Because photobleaching quenches irreversibly only the fluorophores that are in the excited (fluorescent) state, it could be used to separate the fluorescent from the non-fluorescent GLT1-SEP fraction. We therefore implemented a two-photon excitation FRAP protocol in which photobleaching applies

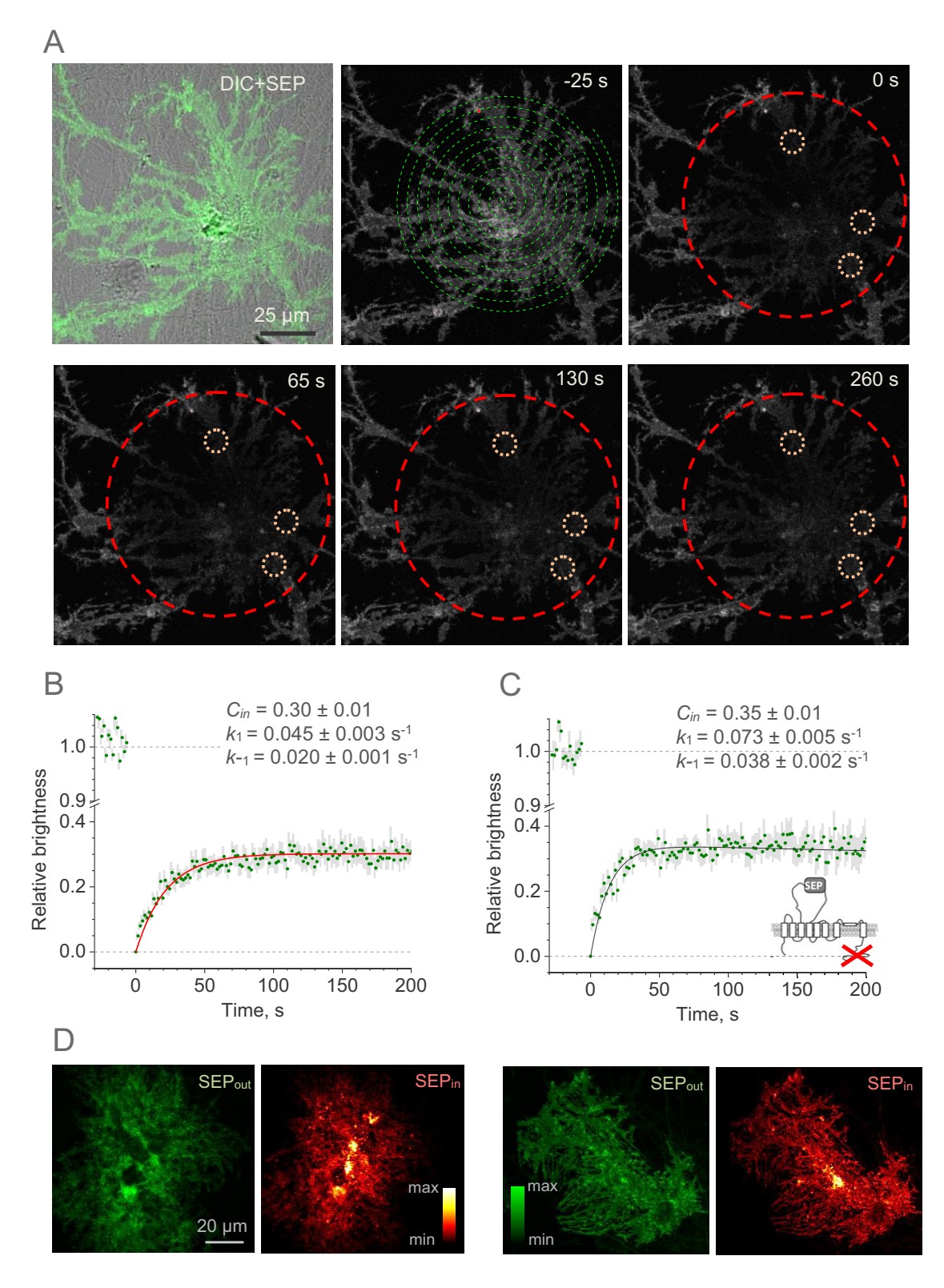

**Figure 2.** Whole-cell FRAP reveals the kinetics of the GLT1-SEP membrane surface turnover. (A) One-cell example illustrating FRAP protocol; upper left, DIC + SEP channel image; serial images, GLT1-SEP channel at different time points (indicated) after a photobleaching pulse ($t = 0$ s); dashed green spiral illustrates application of the tornado laser linescan mode; dashed red circle, laser-photobleached region; dotted orange circles, example of ROIs. (B) Time course (mean ± SEM, n = 27 ROIs in N = 9 cells) of the GLT1-SEP fluorescence intensity within the photobleached region (as in A), normalised

*Figure 2 continued on next page*

*Figure 2 continued*

against the baseline value. Red line, best-fit GLT1-SEP FRAP kinetics incorporating cytosolic protein fraction ($C_{in}$), membrane-surface turnover constants ($k_1$ and $k_{-1}$) and the residual photobleaching constant ($k_b$; not shown); see text and *Figure 2—figure supplement 1* for further detail. (C) Experiment as in (B), but with the with the C-terminus deleted mutant GLT1ΔC-SEP expressed in astroglia (n = 25 ROIs in N = 8 cells); other notations as in (B). (D) Two characteristic examples illustrating cellular distribution of surface-bound fraction of GLT1-SEP (green, SEP$_{out}$) and its intracellular fraction (red, SEP$_{in}$) in live individual astroglia.

The online version of this article includes the following source data and figure supplement(s) for figure 2:

**Source data 1.** Original data readout for *Figure 2B,C*.
**Figure supplement 1.** Establishing the kinetics of whole-cell FRAP for GLT1-SEP molecules in astrocytes.

virtually to the entire astrocyte expressing GLT1-SEP (*Figure 2A*). This was feasible mainly because the morphology of cultured astroglia was essentially two-dimensional, thus permitting comprehensive photobleaching in close proximity of the focal plane. Thus, we used a 'Tornado' laser scan mode, a spiral line-scan that could effectively cover a circular region over just 1–2 ms (*Jensen et al., 2019*), to almost entirely suppress GLT1-SEP fluorescence within the target area (*Figure 2A*, dashed green spiral; area also shown by dashed red circles). This approach enabled us to document partial fluorescence recovery within smaller ROIs inside the bleached area: sampling normally included three ~10 μm wide circular ROIs over cell processes (and additionally one ~20 μm ROI over the soma) in each cell (*Figure 2A*, dotted orange circles). The ROI selection was restricted to morphologically homogenous cell areas inside the fully bleached territory, but otherwise was quasi-random (three ROIs picked randomly out of 10–20 available per cell). These experiments produced the average FRAP time course, with relatively low noise (*Figure 2B*). The cellular biophysical mechanisms underpinning this time course combine membrane insertion of non-bleached GLT1-SEP and, if any, residual photobleaching of surface-bound GLT1-SEP. Solving the corresponding kinetic equations (*Figure 2—figure supplement 1*) provide the resulting fluorescence time course as $C^f_{mem} = R \cdot C_{in}\left(e^{-k_b t} - e^{-k_1 t}\right)$ where $t$ is the time and $k_b$ is the residual photobleaching constant (other notations as above). This equation has two orthogonal (independent) free parameters, $C_{in}$ and $k_1$, whereas the residual photobleaching rate $k_b$ turned out to be negligible throughout the sample. The best-fit estimate gave (*Figure 2B*): $k_1$ = 0.045 ± 0.003 s$^{-1}$, $k_{-1}$ = 0.020 ± 0.001 s$^{-1}$, and $C_{in}$ = 0.30 ± 0.01. Reassuringly, the value of $C_{in}$ (intracellular fraction of GLT1-SEP) obtained in these experiments was indistinguishable from the value of $1-R$ = 0.28 obtained using a fully independent proton permeation method (*Figure 1F*).

These estimates suggest that the characteristic lifetime of the membrane GLT1-SEP fraction, as given by $(k_1)^{-1}$, is ~22 s. Because the cytosolic carboxy-terminal domain of GLT1 has earlier been implicated in the GLT1 expression mechanism (*Gibb et al., 2007*; *Foran et al., 2014*), we asked whether it interferes with the membrane kinetics of the transporter. We therefore expressed GLT1ΔC-SEP in astroglia. FRAP experiments in the GLT1ΔC-SEP expressing cells (*Figure 2C*) showed that deleting the C-terminus had only a moderate effect on the intracellular fraction of transporters ($C_{in}$ = 0.35 ± 0.01) but reduced the GLT1 membrane lifetime by nearly a half (to ~14 s). This finding suggests that the C-terminus could play an important role in retaining GLT1 in the plasma membrane, even though a steady-state membrane-intracellular compartment ratio remains almost unaffected.

Do the cell-average values of the GLT1-SEP membrane fraction (and hence turnover rate) occur homogeneously throughout the cell morphology? To understand this, we directly compared distributions of the membrane and the intracellular populations of GLT1-SEP: the latter was obtained by subtracting the surface GLT1-SEP image from the total GLT1-SEP image (under NH$_4$Cl, as in *Figure 1D*). Intriguingly, this comparison revealed that the membrane GLT1-SEP does not necessarily predict the intracellular GLT1-SEP pattern: the latter could display prominent clustering features (*Figure 2D*). Thus, at least in some cases the membrane dynamics of GLT1 could be specific to microscopic regions of the cell.

## Nanoscale distribution of GLT1 species with respect to synapses

While FRAP measures live GLT1 turnover in the astrocyte membrane, it does not reveal the surface distribution of these molecules, in particular that with respect to synaptic connections. We therefore

turned to super-resolution single-molecule localisation microscopy (SMLM) that involves stochastic localisation of individual molecules (*van de Linde et al., 2011*) using multi-colour 3D SMLM experimental protocols that we have established previously (*Heller et al., 2017*; *Heller and Rusakov, 2019*; *Heller et al., 2020*) (Materials and methods). We thus used chromatically separable photoswitchable dyes to visualise distributions of the native GLT1, GLT1-SEP, or GLT1ΔC-SEP species and their relationship to the synaptic clusters of the ubiquitous postsynaptic density protein PSD95, in mixed cultures (*Figure 3A*, *Figure 3—figure supplement 1A*; as this method used permeabilisation, 25–30% of the label reflected the intracellular fraction of GLT1).

Visualisation with SMLM revealed that the scatter of wild-type GLT1 tends towards forming clusters, both among GLT1 molecules and also between GLT1 and PSD95, and that GLT1-SEP-expressing cells display similar features (*Figure 3B*). Indeed, the classical nearest-neighbour analysis indicated that the pattern of wild-type GLT1 and GLT1-SEP with respect to PSD95 clusters deviates from the evenly random distribution towards closer spatial association (*Figure 3C*), and that these transporter molecules also tend to form short-distance (up to 50 nm) clusters among themselves (*Figure 3—figure supplement 1B*). In contrast, the species with deleted C-terminus, GLT1ΔC-SEP, showed spatial dissociation (distancing) with PSD95 clusters (*Figure 3B,C*) while displaying dense molecular clustering among themselves, to the extent that the latter is not distinguishable from uniform packing at a high local density (*Figure 3—figure supplement 1B*; this analysis does not cover higher-order, longer-distance GLT1ΔC-SEP clustering, which is evident in *Figure 3B*). These observations indicate that the C-terminus of GLT1 plays a critical role not only in its cellular membrane turnover but also in the surface (and submembrane) expression pattern of the protein.

## Lateral mobility of GLT1-SEP in astroglia

We next set out to assess lateral surface mobility of GLT1-SEP using a classical FRAP protocol, in which the fluorescence kinetics is monitored within a small ROI (*Figure 4A*). Because running a FRAP protocol bleaches immobile molecules that remain within the ROI, repeating this protocol within the same ROI may produce a different FRAP time course. To account for this and any other use-dependent trends in the imaging conditions, we routinely recorded pairs of FRAP trials separated by 1 min (*Figure 4B*, *Figure 4—figure supplement 1*), unless indicated otherwise. This time interval was also longer than the GLT1 membrane turnover period (~22 s, see above), which should help minimise the number of bleached immobile molecules remaining within the ROI, as they are replaced by new arrivals from the intracellular compartment. This approach enabled us to compare FRAP kinetics between control conditions and during ligand application, in the manner that provides correction for any consistent difference within paired FRAP trials (*Figure 4C*, *Figure 4—figure supplement 1*).

In the first experiment, we therefore documented FRAP kinetics within a small (~1.6 µm diameter) circular membrane area of a visualised astrocyte, in baseline conditions and during a brief (250 ms, 1 mM) application of glutamate 200 ms prior to bleaching onset (*Figure 4C*, right panel), to mimic a transient rise in local excitatory activity.

The data (corrected for paired-trial trends) showed a clear difference in the FRAP kinetics between the two conditions (*Figure 4D*, left). This could reflect a difference in lateral diffusivity of mobile transporters, but also in the immobile versus mobile fractions of GLT1-SEP. To evaluate both variables from the FRAP kinetics, we used the well-established Soumpasis approach for circular ROIs (*Soumpasis, 1983*; *Kang et al., 2009*) (Materials and methods). This fitting method operates with only two mutually independent (orthogonal) free parameters, mobile fraction $C_{mob}$, and diffusion coefficient $D$, and its estimates should not depend on residual changes in fluorescence, such as photobleaching (*Soumpasis, 1983*; *Kang et al., 2009*).

In baseline conditions, the best-fit values were $C_{mob}$ = 0.76 ± 0.01 and mobile-fraction diffusivity $D$ = 0.152 µm²/s (diffusion time $\tau_D$ = 1.75 ± 0.03 s, Materials and methods), thus giving the average diffusivity (accounting for mobile and immobile molecules) $D^*$ = $C_{mob} \cdot D$ = 0.114 µm²/s (*Figure 4D*, right). This value appears in correspondence with the average lateral diffusivity of GLT1 measured earlier with QDs (*Murphy-Royal et al., 2015*), although it is higher than the values reported using a different QD approach (*Al Awabdh et al., 2016*). When glutamate was briefly applied immediately before and after the photobleaching pulse, diffusivity of the mobile-fraction only did not appear to be affected ($\tau_D$ = 1.73 ± 0.03 s) whereas its size increased significantly ($C_{mob}$ = 0.88 ± 0.003), giving average $D^*$ = 0.135 µm²/s, an increase of ~18% compared to control (*Figure 4D*, right). This result suggested that glutamatergic activity could boost overall membrane mobility of GLT1 transporters,

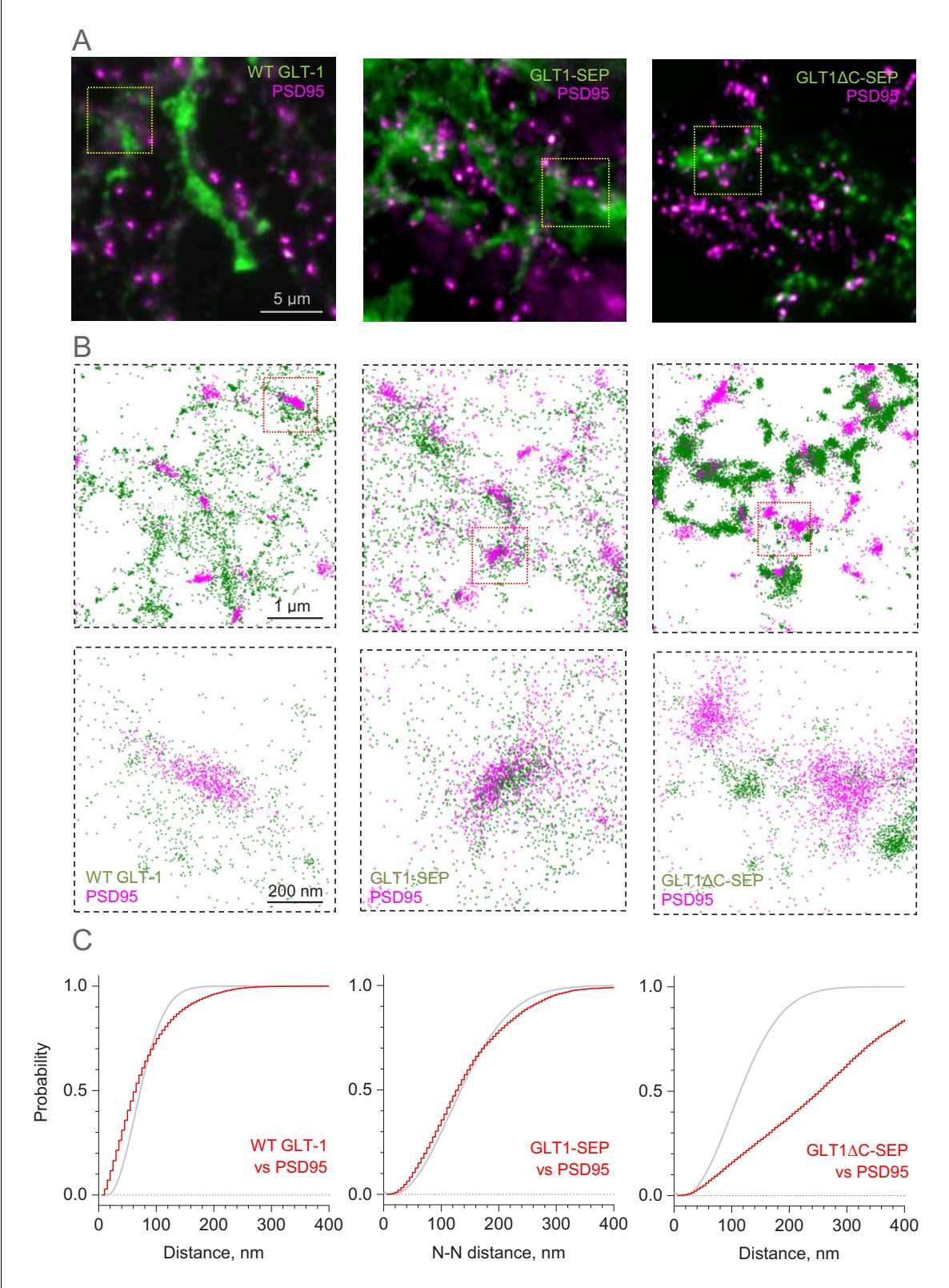

**Figure 3.** Distribution of GLT-1 species in relation to postsynaptic densities in the astroglial membrane: A super-resolution SMLM analysis. (A) Wide-field fluorescent images (examples) illustrating antibody labelled GLT1 species (green channel) and postsynaptic density protein PSD95 (magenta), as indicated, in mixed astroglia-neuron cultures. See *Figure 3—figure supplement 1A* for macroscopic views. (B) SMLM nano-localisation maps (examples) depicting individual labelled GLT1 species (green), as indicated, and PSD95 (magenta) molecules. Top row, ROIs shown as the

*Figure 3 continued on next page*

Figure 3 continued

corresponding yellow squares in (**A**); bottom row, ROIs shown as red squares in the top row. (**C**) Red line (5 nm bins): distribution $D(r)$ of nearest-neighbour (N–N) distances $r$ between labelled GLT1 species and clusters of PSD95 molecules (PSD95 clusters represent >50 particles <100 nm apart). Grey line: theoretical distribution $D(r) = 1 - exp(-\lambda \pi r^2)$ that corresponds to the Poisson point process (evenly random scatter) with the same surface density of PSD95 clusters $\lambda$ as sampled experimentally. Experimental $\lambda$ values were: 67 $\mu m^{-2}$ (WT GLT-1), 15.5 $\mu m^{-2}$ (GLT1-SEP), and 22.2 $\mu m^{-2}$ (GLT1Δ-SEP); see Materials and methods for details.

The online version of this article includes the following source data and figure supplement(s) for figure 3:

**Source data 1.** Original data readout for *Figure 3C* and *Figure 3—figure supplement 1B*.

**Figure supplement 1.** Distribution of GLT-1 species in the astroglial membrane: macroscopic wide-field view and super-resolution SMLM analysis.

a conclusion similar to that drawn earlier using QDs (*Murphy-Royal et al., 2015*; *Al Awabdh et al., 2016*).

## Molecular regulators of activity-dependent membrane mobility of GLT1

We next found that deleting the C-terminus of GLT1-SEP does not alter its mobility in basal conditions ($C_{mob}$ = 0.815 ± 0.003; $D\star$ = 0.117 $\mu m^2$/s) but appears to block the mobility-boosting effect of glutamate application (*Figure 4E*). A similar result was obtained when metabotropic glutamate receptors were blocked by a pharmacological cocktail: no detectable effect on GLT1-SEP mobility in baseline conditions ($C_{mob}$ = 0.795 ± 0.003; $D\star$ = 0.121 $\mu m^2$/s) but suppression of the glutamate-induced mobility increase (*Figure 4F*). Because purinergic receptors mediate a major signalling cascade in brain astroglia (*Verkhratsky and Nedergaard, 2018*), we asked whether ATP application alters mobility of GLT1-SEP and detected no effect (*Figure 4G*).

Finally, to assess sensitivity and the dynamic range of our FRAP protocol we cross-linked surface GLT-SEP, by incubating cultures briefly (10 min in humidified incubator) with either IgY antibody (100 µg/ml, chicken polyclonal, Merck AC146) or with the anti-GFP antibody (100 µg/ml, chicken polyclonal, Abcam ab13970). The cross-linkage reduced the FRAP-measured transporter mobility fivefold (*Figure 4H*), confirming high sensitivity and general suitability of the present FRAP method.

## Cellular mechanisms affecting GLT1 mobility in hippocampal slices

Whilst cultured astroglia are thought to retain key molecular mechanisms acting in situ, astrocytes in organised brain tissue have distinct morphology and engage in network signalling exchange that may be different from cultures. We therefore set out to validate our key observations focusing on area CA1 astroglia in organotypic hippocampal slices: these cells closely resemble their counterparts in vivo (*Figure 5A*) and are embedded in a well-defined synaptic circuitry. To induce a rapid rise in the spontaneous excitatory activity of the native network, we blocked GABA$_A$ receptors and potassium channels with bicuculline and 4-AP (rather than applying glutamate, *Figure 5B*).

Because the morphology of astroglia in brain tissue is essentially three-dimensional, the whole-cell FRAP protocols (as in *Figure 2A*) were not technically feasible. However, the small-ROI FRAP experiments (as in *Figure 4A*) in slices showed that diffusivity of GLT1-SEP in the plasma membrane was similar, if somewhat slower, than that in cultures (*Figure 5C*). Elevated excitatory activity increased GLT1-SEP mobility, which could be reversed by blocking spiking activity with TTX (*Figure 5C*). We confirmed that this effect was not due to some unknown concomitants of increased network activity that might affect astrocyte membrane properties per se: a truncated sham-protein probe carrying an extracellular SEP domain showed no changes in lateral diffusion under this protocol (*Figure 5—figure supplement 1A*). Conversely, application of the vehicle DMSO on its own had no effect on the activity-dependent increase in GLT1-SEP diffusion (*Figure 5—figure supplement 1B*).

Again, deletion of the C-terminus or the pharmacological blockade of metabotropic glutamate receptors suppressed the activity-dependent mobility increase (*Figure 5D,E*). Because metabotropic glutamate receptors engage a major $Ca^{2+}$ signalling cascade in astroglia (*Porter and McCarthy, 1997*), we asked if buffering intracellular $Ca^{2+}$ with BAPTA-AM is involved and found this to be the case (*Figure 5F*). Investigating this further, we blocked the calcium and calmodulin-dependent phosphatase calcineurin, which produced similar suppression (*Figure 5G*). However, non-selective protein

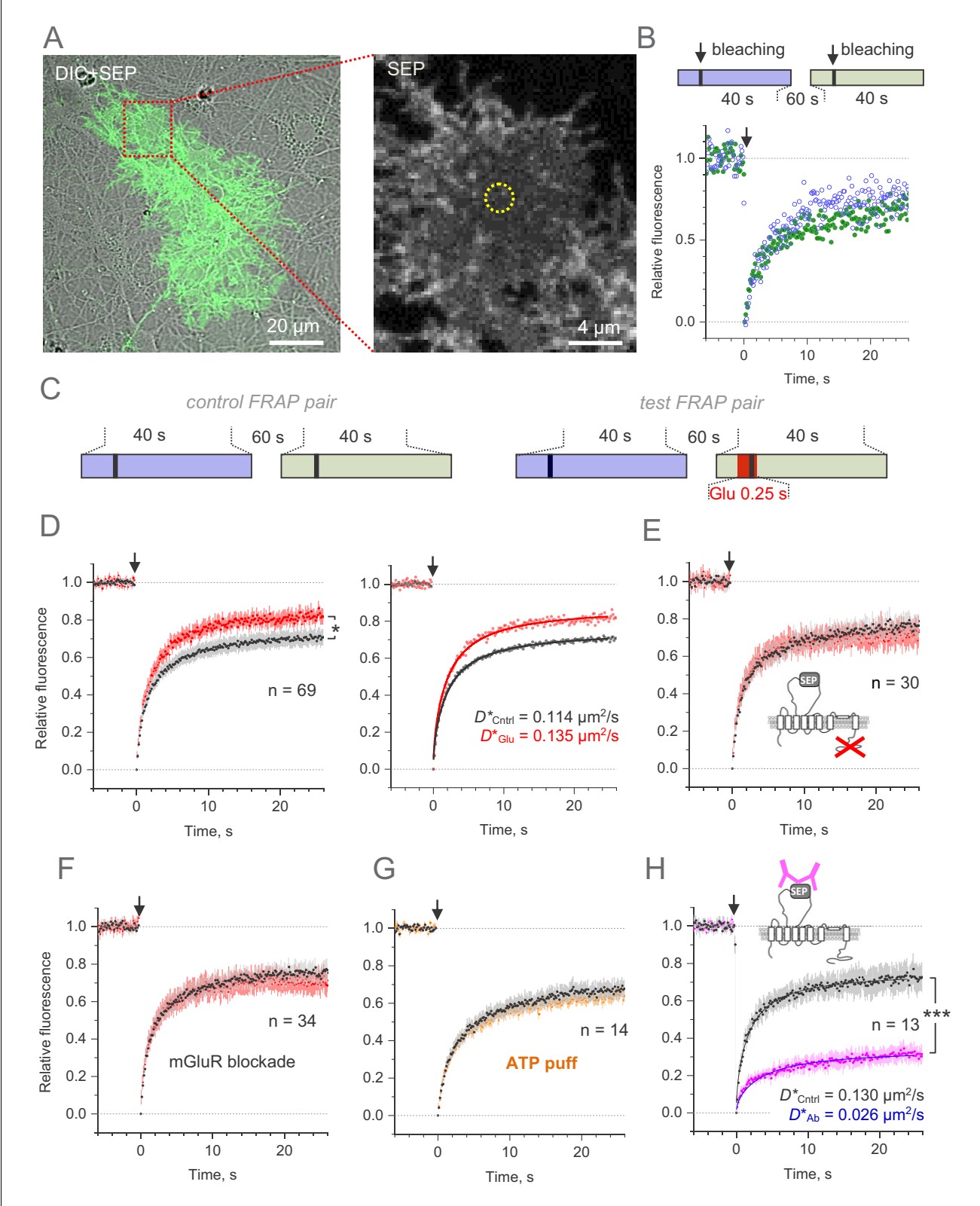

**Figure 4.** Microscopic-ROI FRAP probes lateral membrane mobility of GLT1-SEP in cultured astroglia. (**A**) One-cell example as seen in GLT1-SEP + DIC channel (left), with a selected area (dotted rectangle) illustrating a circular, 2.06 µm wide FRAP spot (dotted circle, right). (**B**) Diagram, the paired-sample FRAP protocol, in which two trials are carried out in succession, to account for any non-specific, time-dependent drift in FRAP kinetics. Plots, one-cell example of the paired-sample FRAP test, with the first and second trial data are shown in blue and green, respectively; arrow, bleaching pulse

*Figure 4 continued on next page*

*Figure 4 continued*

($\lambda_x^{2P}$ = 690 nm, 10–15 mW under the objective, duration 46 ms); fluorescence ROI, photobleaching spot as in (**A**). (**C**) Diagram illustrating the paired-sample FRAP protocol, which includes both control and glutamate application cycles; FRAP kinetics under glutamate application could be corrected for non-specific drift by using the control cycle data. (**D**) *Left*, average time course of the GLT1-SEP FRAP (dots and shade: mean ± 95% confidence interval, here and thereafter) in baseline conditions (black) and upon glutamate application (250 ms puff 200 ms before the photobleaching pulse lured); asterisk, p < 0.05 (n = 69 FRAP spots in N = 13 cells). *Right*, FRAP time course (mean values) fitted with the Soumpasis FRAP equation for (see main text) for control and glutamate tests. Best-fit GLT1-SEP diffusion coefficient *D* is shown for control (Cntrl) and glutamate puff (Glu) trials, as indicated. (**E**) Average FRAP time course in control and glutamate-puff tests carried out with the C-terminus deleted mutant CLT1ΔC-SEP, as indicated (n = 30 FRAP spots in N = 7 cells); other notation as in (**D**). (**F**) Average FRAP time course in control and glutamate-puff tests in the presence of AMPA and metabotropic glutamate receptor (mGluR) blockers (n = 34 FRAP spots in N = 7 cells): MPEP (1 mM), LY341495 (30 nM), YM298198 (0.3 μM); NBQX (10 μM) was added to suppress network hyper-excitability under LY341495; other notation as in (**D**). (**G**) Average FRAP time course in control conditions and after the ATP pressure puff (100 μM, 250 ms duration 200 ms before bleaching start, no glutamate), as indicated (n = 14 FRAP spots in N = 4 cells); other notation as in (**D**). (**H**) Control test: Average FRAP time course in control conditions and under surface cross-linkage by anti-GFP antibody, as indicated (n = 13 FRAP spots in N = 2 cells); other notation as in (**D**).

The online version of this article includes the following source data and figure supplement(s) for figure 4:

**Source data 1.** Original data readout for *Figure 4B,D–H*.

**Figure supplement 1.** Microscopic-ROI FRAP probes lateral membrane mobility of GLT1-SEP in cultured astroglia.

kinase inhibition with the antibiotic staurosporine left the excitation-induced rise of GLT1-SEP mobility intact (*Figure 5H*), thus narrowing the range of the candidate molecular mechanisms involved.

Throughout these experiments, we compared GLT-1 mobility before and after 4-AP application, under exactly the same pharmacological challenge. At the same time, the pharmacological manipulations per se had no detectable influence on the prominent boosting effect of 4-AP on network activity (as in *Figure 5B*). We therefore considered that, in the present experimental design, any concomitant effects of drug application on neurons were largely cancelled out, for comparison purposes.

## Discussion

Here, we developed a functional fluorescent analogue of the main glial glutamate transporter GLT1, termed GLT1-SEP, and used it to evaluate its membrane dynamics, incorporating both surface mobility and membrane-intracellular compartment turnover in astrocytes. We used patch-clamp electrophysiology, immunostaining assays, and super-resolution SMLM imaging to confirm that glutamate transport properties of GLT1-SEP and its cell expression are generally compatible with its wild-type counterpart. Clearly, virtually any direct experimental intervention, such as patch-clamp, genetically encoded sensors, opsins, or fluorescent tags, by definition introduce some perturbation to the system, which is difficult to control entirely. However, FRAP applied to fluorophore-fused proteins has for decades proved a key method to grasp accurately their membrane kinetics, and fluorescent tagging has been successfully implemented in recent protein turnover studies (*Rudolph et al., 2019*). Our live-imaging and immunostaining data also indicate that the expression of GLT1-SEP was consistently dense and relatively even across the astrocyte surface, similar to that of native GLT1, which argues for physiological relevance of our approach.

Taking advantage of the pH-sensitive fluorescence and photobleaching properties of GLT1-SEP, we established that the 70–75% fraction of its cellular content reside on the astrocyte surface, with a characteristic turnover rate of 0.04–0.05 s$^{-1}$, which corresponds to a 20–25 s cycle. The fact that the population of functional astroglial glutamate transporters is effectively replaced several times per minute must be an intriguing discovery. Interestingly, a recent study used fixed-tissue immunocytochemistry in pure astroglial cultures to find only ~25% of all GLT1 expressed in the cell membrane (*Underhill et al., 2015*), thus relating a boost in transporter numbers to the presence of neuronal connections (in the mixed cultures slices employed here).

Removing the C-terminus of GLT1-SEP only moderately increased its intracellular fraction while substantially reducing its plasma membrane lifetime. These observations suggest an important contribution of the C-terminus to the retaining of GLT1 molecules on the astrocyte surface. Intriguingly, SMLM imaging revealed that deletion of the C-terminus severely disrupts the cell surface pattern of

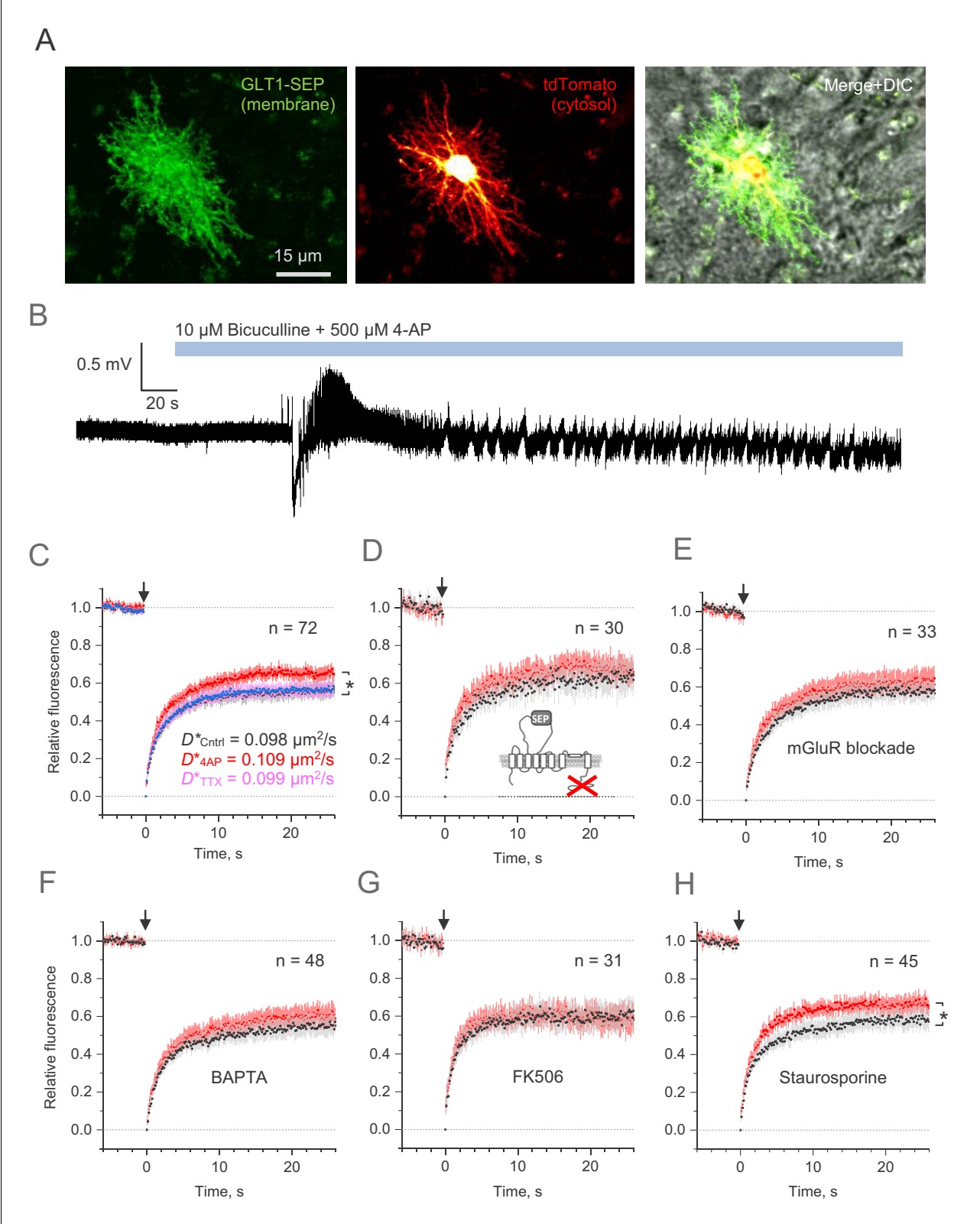

**Figure 5.** Microscopic-ROI FRAP probes lateral membrane mobility of astroglial GLT1-SEP in organotypic hippocampal slices. (**A**) Example of astroglia in an organotypic slice, seen in GLT1-SEP, tdTomato, and merge +DIC channel, as indicated. (**B**) One-slice example of boosted excitatory activity (field potential recording, CA1 area) induced by the application of GABA$_A$ receptor blocker Bicuculine and the potassium channel blocker 4-AP, as indicated. (**C**) Average time course of the GLT1-SEP FRAP (dots and shade: mean ± 95% confidence interval, here and thereafter) in baseline conditions (black),

*Figure 5 continued on next page*

**Figure 5 continued**

Bicuculine + 4-AP application (red), and after sodium channel blockade by TTX (magenta), as indicated; p < 0.05 (n = 72 FRAP spots in N = 15 cells). *Right*, FRAP time course (mean values) fitted with the Soumpasis FRAP equation for (see main text) for control and glutamate tests. Best-fit GLT1-SEP diffusion coefficient $D*$ (Soumpasis FRAP fit) is shown for control (Cntrl), Bicuculine + 4-AP application (4AP) and TTX trials, as indicated. (D) Average FRAP time course for the C-terminus deleted mutant GLT1ΔC-SEP, as indicated; other notation as in (C). (E) Average FRAP time course in the presence of metabotropic glutamate receptor blockers (n = 33 FRAP spots in N = 8 cells): MPEP (1 μM), LY341495 (30 nM), YM298198 (0.3 μM), and NBQX (10 μM); other notation as in (C). (F) Average FRAP time course in the presence of intracellular BAPTA (n = 48 FRAP spots in N = 10 cells); other notation as in (C). (G) Average FRAP time course under the calcineurin (phosphatase) blockade by FK506 (1 μM; n = 31 FRAP spots in N = 6 cells); other notation as in (C). (H) Average FRAP time course in the presence of the broad-range kinase activity blocker Staurospotine (100 nM); *p < 0.05 (n = 45 FRAP spots in N = 8 cells) other notation as in (C).

The online version of this article includes the following source data and figure supplement(s) for figure 5:

**Source data 1.** Original data readout for *Figure 5C–H* and *Figure 5—figure supplement 1A,B*.

**Figure supplement 1.** Control tests for microscopic-ROI FRAP probing of lateral membrane mobility of GLT1-SEP in organotypic hippocampal slices.

GLT1 and its spatial relationship with neighbouring synaptic connections (represented by clusters of PSD95).

It has previously been shown that GLT1 is endocytosed constitutively, in a clathrin-dependent manner, taking the transporter into rapidly recycling endosomes containing EEA1 and Rab4 (*Martínez-Villarreal et al., 2012*). Earlier studies have also indicated that the common neuronal glutamate transporter GLAST also undergoes clathrin-dependent endocytosis (*González et al., 2007*). Using reversible biotinylation followed by immunocytochemistry, Robinson's group obtained estimates of the membrane residence time of EEAT1 (*Fournier et al., 2004*), and subsequent studies identified several molecular cascades that control cell surface expression of GLAST and GLT1 including ubiquitination and sumoylation (*González et al., 2007*; *García-Tardón et al., 2012*; *Martínez-Villarreal et al., 2012*; *Piniella et al., 2018*). While the biochemical machinery of GLT1 turnover is outside the scope of the present study, its investigation should provide further insights into the adaptive features of glutamate transport in the brain.

We next employed GLT1-SEP to investigate its lateral mobility in the plasma membrane, and the regulatory mechanisms involved. A similar question has been elegantly explored in two studies using single-particle tracking with QDs (*Murphy-Royal et al., 2015*; *Al Awabdh et al., 2016*). However, the key advantage of the present approach is that it accounts for membrane-intracellular compartment exchange, in addition to lateral mobility per se: tracking QD-labelled GLT1 must ignore the non-labelled GLT1 fraction that is being constantly delivered to the cell surface. We found relatively high average lateral diffusivity (0.10–0.15 μm$^2$/s), but also a significant fraction of immobile transporters (25–30%). Importantly, the characteristic lateral diffusion time of the GLT1-SEP mobile fraction (~1.75 s) was much shorter than its membrane lifetime of ~22 s. This implies that the assessment of mobile transporter diffusivity, obtained here with GLT1-SEP or earlier with QDs, should not be noticeably influenced by its membrane turnover. Nonetheless, the latter could have a critical effect on the dynamics of the immobile (slowly moving) fraction of GLT1. For instance, the earlier studies found that GLT1 near synapses diffuse orders of magnitude slower than all transporters on average (*Murphy-Royal et al., 2015*; *Al Awabdh et al., 2016*). Thus, the membrane-intracellular compartment exchange, rather than lateral diffusion, could be a preferred mechanism of the transporter turnover near synapses. Indeed, it has been estimated that small excitatory synapses release ~3000 glutamate molecules per vesicle (*Savtchenko et al., 2013*) and that 10,000–15,000 GLT1 molecules are present within 1 μm$^3$ of CA1 hippocampal neuropil (*Lehre and Danbolt, 1998*). Therefore, a rapid burst of several single-vesicle release events, or repeated multi-vesicular release, could lead to a relatively high occupancy, if not saturation, of immobile GLT1 within the 1 μm$^3$ perisynaptic volume. The latter highlights the importance of reliable local resupply of GLT1, arguably through membrane turnover.

The present method has its own limitations. Similar to the QD approach, or any other live molecular tagging method, it is not technically feasible to verify fully that the labelled (or mutated) molecules have exactly the same dynamic properties as their native counterparts. Nonetheless, it is reassuring that the average lateral mobility of GLT1-SEP found here was similar to that estimated using QDs (*Murphy-Royal et al., 2015*), despite two very different modes of interference with the molecular structure.

The potential importance of high GLT1 diffusivity for regulating the waveform of excitatory synaptic currents was suggested earlier (*Murphy-Royal et al., 2015*). This might indeed be the case for large synapses, with multiple release sites (*DiGregorio et al., 2002*), that are prevalent in cultures or incubated slices. At small central synapses in situ, however, the kinetics of individual AMPA currents should not depend on glutamate buffering outside the synaptic cleft (*Zheng et al., 2008*; *Savtchenko et al., 2013*). Nonetheless, intense glutamatergic activity can boost glutamate escape from the cleft (*Lozovaya et al., 1999*), in which case lateral movement of astroglial transporters could indeed contribute to the efficiency of uptake.

Our results should provide critical real-time turnover data complementing the well-explored cellular machinery of GLT1 exocytosis and recycling in the plasma membrane (*González et al., 2007*; *García-Tardón et al., 2012*; *Martínez-Villarreal et al., 2012*; *Piniella et al., 2018*). At the same time, mechanisms that control lateral diffusion of GLT1 on the astroglial surface are only beginning to transpire. Two previous studies detected a diffusion-facilitating role of glutamate, which was either applied exogenously or released through intense neuronal network activity (*Murphy-Royal et al., 2015*; *Al Awabdh et al., 2016*), suggesting an adaptive function of GLT1 mobility. Our results confirm these observations, but also provide further important functional associations between the expected sources of molecular signalling in the brain and GLT1 mobility. We found that the deletion of the C-terminus, or the blockade of glutamate receptors, intracellular $Ca^{2+}$ buffering, or the suppression of the calcium and calmodulin-dependent phosphatase calcineurin made the GLT1 membrane mobility irresponsive to glutamate. This is in line with previous studies which have shown that blocking kinase activity promotes glutamate uptake (*Adolph et al., 2007*; *Li et al., 2015*): lateral mobility might be one of the mechanisms assisting this process. Although regulation of GLT1 by calcineurin has previously been shown on the transcriptional level (*Sompol et al., 2017*), calcineurin is also known to directly dephosphorylate membrane proteins such as connexin-43 (*Tencé et al., 2012*). At the same time, ATP application (which triggers prominent $Ca^{2+}$-dependent cascades in astrocytes) had no effect on GLT1 mobility. We have thus identified several molecular signalling cascades that might provide important clues to the possible regulatory intervention in brain pathologies associated with malfunctioning astroglial glutamate uptake (*Fontana, 2015*; *Peterson and Binder, 2019*).

## Materials and methods

DNA constructs cDNA of rat GLT1 cloned by Baruch Kanner group (*Pines et al., 1992*) under CMV promoter was a generous gift from Michael Robinson. Superecliptic pH-luorin (SEP) was introduced into second intracellular loop of GLT1 using standard cloning techniques. First, GLT1 sequence was mutated with QuikChange II Site-Directed Mutagenesis Kit (Agilent) using the following pair of primers: GTTCTGGTGGCACCTACGCGTCCATCCGAGGAG and CTCCTCGGATGGACGCGTAGGTGCCACCAGAAC in order to introduce MluI restriction site. Subsequently, SEP was amplified using pair of primers: CCGGACGCGTCTGGTTCCTCGTGGATCCGGAGGAATGAGTAAAGGAGAAGAACTTTTCAC and CCGGACGCGTTCCAGAAGTGGAACCAGATCCTCCTTTGTATAGTTCATCCATGCCATG, which introduced linkers and enabled subcloning SEP into MluI restriction site. Resulting GLT1-SEP was subcloned into pZac2.1 gfaABC1D-tdTomato (Addgene Plasmid #44332) (*Shigetomi et al., 2013*) using BmtI and XbaI sites in order to be expressed under glia-specific gfaABC₁D promoter. GLT1ΔC-SEP was generated using the following pair of primers: CCGATCTCGAGATGGCATCAACCGAGGGTG and CCGATGGTACCCTAGACACACTGATTAGAGTTGCTTTC which introduces 'amber' stop codon after Val537 in GLT1 sequence. GLT1ΔC-SEP was then cloned to plasmid pZac2.1gfaABC1D_MCS which was generated by replacing tdTomato in pZac2.1 gfaABC1D-tdTomato with hybridised pair of oligonucleotides: AATTCACCGGTGGCGCGCCGGATCCTGTACAACGCGTGATATCGGTACCCATATGCCGCGGACTAGTT and CTAGAACTAGTCCGCGGCATATGGGTACCGATATCACGCGTTGTACAGGATCCGGCGCGCCACCGGTG cloned into EcoRI and XbaI sites. eGFP-GLT1 was generated by amplification of GFP with the following pair of primers: CTATAGGCTAGCATGGTGAGCAAGGGCG and CGTAACTCGAGGAATTCGCCAGAACCAGCAGCGGAGCCAGCGGATCCCTTGTACAGCTCGTCCATG which introduced linker at 3' end of GFP and enabled for it using BmtI and XhoI sites at 5' end of GLT1 in pCMV_GLT1 plasmid. Resulting eGFP-GLT1 was subcloned into pZac2.1gfaABC1D_MCS using BmtI and XbaI restriction sites. pDisplay-SEP was generated by subcloning SEP, amplified with a pair of primers: CCGCGAAGATCTATGAG

TAAAGGAGAAGAACTTTTCAC and GGCAGTCGACCTGCAGCCGCGGCCGTTTGTATAGTTCA TCCATGCCATG into pDisplay-mSA-EGFP-TM (Addgene plasmid #39863) (*Lim et al., 2013*) using BglII and SalI restriction sites.

## Cell cultures

HEK 293T (Lenti-X 293T subclone, TaKaRa) were maintained in DMEM, high glucose, GlutaMAX (Thermo Fisher Scientific) supplemented with 10% foetal bovine serum (FBS; Thermo Fisher Scientific). For transfection and patch-clamp experiments cells were plated at density 25,000 cells per 13-mm-diameter coverslip (Assistent, Germany) coated with poly-L-lysine (Sigma-Aldrich). Cells were co-transfected with plasmids coding GLT1 or GLT1-SEP under CMV promoter together with mRFP1 under β-actin promoter in a 2:1 ratio using Lipofectamine 2000 (Thermo Fisher Scientific) according to the manufacturer's instructions. Transfected cells were used for patch clamp experiments the next day.

## Electrophysiology

Patch clamp recordings were made from transfected HEK cells. Coverslips with cells were perfused with extracellular solution containing 125 mM NaCl, 2.5 mM KCl, 2 mM $CaCl_2$, 1.3 mM $MgSO_4$, 26 mM $NaHCO_3$, 1.25 mM $NaH_2PO_4$, 12 mM D-glucose, bubbled with 95:5 $O_2/CO_2$ (pH 7.4). Patch pipettes were pulled to resistance of 4–5 MOhm when filled with the intracellular solution containing 120 mM CsCl, 8 mM NaCl, 10 mM HEPES, 0.2 mM $MgCl_2$, 2 mM EGTA, 2 mM MgATP, 0.3 mM $Na_3GTP$ (pH 7.3). Cells were voltage-clamped at –70 mV, recordings were performed at 33–35°C, and signals digitised at 10 kHz. For glutamate application, we used a θ-glass pipette pulled out to an ~200 µm tip diameter, as described and illustrated earlier (*Sylantyev and Rusakov, 2013*). Briefly, a thin capillary was inserted into each θ-glass channel, connected to the two-channel PDES-2DX-LA pneumatic microejector (npi electronic GmbH), with pressure adjusted using compressed nitrogen. The θ-glass pipette was attached to a Bender piezoelectric actuator (PL127.11, Physik Instrumente) mounted on a micro-electrode holder held in a precision micromanipulator with remote control (Scientifica Ltd). Square electric pulses of chosen duration were applied to the actuator using a constant-voltage stimulus isolator (DS2, Digitimer), which moved the θ-glass pipette so that the tested cell was exposed to either channels, with the channel boundary monitored in the DIC channel (*Sylantyev and Rusakov, 2013*).

## Primary dissociated culture

Dissociated hippocampal cultures from P0 (postnatal day 0) Sprague-Dawley rats were prepared in full compliance with the national guideline and the European Communities Council Directive of November 1986, and the European Directive 2010/63/EU on the Protection of Animals used for Scientific Purposes. Brains were removed and hippocampi were isolated on ice in dissociation medium – DM (81.8 mM $Na_2SO_4$, 30 mM $K_2SO_4$, 5.8 mM $MgCl_2$, 0.25 mM $CaCl_2$, 1 mM HEPES pH 7.4, 20 mM glucose, 1 mM kynureic acid, 0.001% phenol red), hippocampi were later incubated twice for 15 min at 37°C with 100 units of papain (Worthington, NY) in DM and rinsed three times in DM and subsequently three times in plating medium (MEM, 10% FBS and 1% penicilin–streptomycin; Thermo Fisher Scientific). Hippocampi were triturated in plating medium until no clumps were visible and cells were diluted 1:10 in OptiMEM (Thermo Fisher Scientific), centrifuged for 10 min at room temperature (RT), at 200 × g. The resulting cell pellet was suspended in plating medium, cells were counted in 1:1 dilution of 0.4% Tryptan Blue solution (Thermo Fisher Scientific) and plated at density 75,000 cells per 13-mm-diameter coverslip (Assistent, Germany) coated with 1 mg/ml poly-DL-lysine (Sigma-Aldrich, P9011) and 2.5 µg/ml laminin (Sigma-Aldrich, L2020). After 3 hr plating medium was exchanged for maintenance medium (Neurobasal-A without phenol red, 2% B-27 supplement, 1% penicillin–streptomycin, 0.5 mM glutaMAX, 25 µM β-mercaptoethanol; Thermo Fisher Scientific) and cells were kept at 37°C, under a humidified 5% $CO_2$ atmosphere. Cells were transfected with plasmids using Lipofectamine 3000 (Thermo Fisher Scientific) at 7–10 days in vitro (DIV). Lipofectamine–DNA complexes were prepared according to the manufacturer's instructions and were incubated with cells for 1 hr in the incubator, in fresh transfection medium (MEM without phenol red, 2% B27 supplement, 1 mM pyruvate, 0.5 mM GlutaMAX, 25 µM β-mercaptoethanol; Thermo Fisher

Scientific). After transfection conditioned maintenance medium was returned to cells. All experiments were performed at 14–19 DIV.

## Western blot in HEK cells

HEK cells were co-transfected with plasmids coding GLT1 or GLT1-SEP under CMV promoter together with mRFP1 under β-actin promoter in a 2:1 ratio using Lipofectamine 2000 (Thermo Fisher Scientific), according to the manufacturer's instructions. After 24 h cells were lysed using $1\times$ Leammli solution, and the samples were loaded on 10% polyacrylamide gels. The samples were electro-transferred onto polyvinylidene difluoride membranes (Immobilon-P, Millipore), which were blocked 2 hr at RT with 10% nonfat milk in Tris-buffered saline with 0.1% Tween 20 (TBS-T). After blocking, the membranes were incubated at 4℃ overnight with the following antibodies: 1:1000 anti-GLT1(ex) (Synaptic Systems # 250 203; AB11042312), diluted in 5% non-fat milk in TBS-T. The membranes were next incubated for 2 hr at RT with the peroxidase-labelled secondary antibody (Goat Anti-Rabbit IgG Antibody; Vector Laboratories, #PI-1000; AB_2336198) diluted 1:10,000 in 5% non-fat milk in TBS-T. After washing, peroxidase activity was visualised with the ECL Prime reagent (GE Healthcare #RPN2232). The membranes were next stripped for 1 hr at 50℃ with a buffer containing: 62.5 mM Tris-Cl pH 6.8, 2% SDS, and 120 mM ß-mercaptoethanol, washed three times with TBS-T, blocked for 2 hr at RT with 10% non-fat milk in TBS-T, and probed with anti-alpha-Tubulin antibody (Sigma-Aldrich #T9026; AB_477593) diluted 1:5000 in 5% non-fat milk. The membranes were washed three times with TBS-T and incubated with peroxidase-labelled secondary antibody (Horse Anti-Mouse IgG Antibody, Vector Laboratories, #PI-2000; AB_2336177) diluted 1:10,000 in 5% non-fat milk. After washing, peroxidase activity was visualised with the ECL Prime reagent.

## Immunostaining in HEK cells

HEK cells were co-transfected with the plasmids coding GLT1 or GLT1-SEP under CMV promoter, together with mRFP1 under β-actin promoter, in a 2:1 ratio using Lipofectamine 2000 (Thermo Fisher Scientific), according to the manufacturer's instructions. After 24 hr the cells were fixed using 37℃ pre-warmed 4% paraformaldehyde in PBS for 10 min at 37℃, were washed thrice in PBS, permeabilised in 0.1% Triton X-100 for 10 min, and blocked with 3% BSA and 10% normal goat serum in PBS for 1 hr at RT. Afterwards, cells were incubated with the primary antibody (see below) in 1.5% BSA and 5% normal goat serum in PBS overnight at 4℃, washed thrice with PBS, incubated with the secondary antibody (see below) in 1.5% BSA and 5% normal goat serum in PBS for 2 hr, washed thrice with PBS. The cells were mounted using Fluoromount-G Mounting Medium (Thermo Fisher Scientific #00-4958-02) and stored at 4℃ until imaging.

The primary antibodies used were: glial glutamate transporter GLT1 (guinea pig, polyclonal, synthetic peptide from the C-terminus of rat GLT1, Merck, AB1783, AB_90949, dilution 1:1,000) and GFP (chicken, polyclonal, GFP directly from Aequorea Victoria, Thermo Fisher Scientific, A10262, AB_2534023, dilution 1:1,000). The secondary antibodies: anti-chicken IgY (goat, Alexa Fluor 488, Thermo Fisher Scientific, A32931, dilution: 1:1,000) and anti-guinea pig IgG (goat, Alexa Fluor 647, Thermo Fisher Scientific, A-21450, dilution: 1:1,000). Images were recorded with Zeiss 780 confocal microscope using 40×/1.4 NA Oil Plan-Apochromat objective. As HEK cells do not express endogenous GLT1, only the cells which transfected with mRFP1 were analysed for GLT1 expression.

## Organotypic hippocampal culture

Transverse hippocampal organotypic cultures were prepared according to Stoppini and colleagues (*Stoppini et al., 1991*) with some modifications. P8 Sprague-Dawley rats were sacrificed in full compliance with the national guideline and the European Communities Council Directive of November 1986, and the European Directive 2010/63/EU on the Protection of Animals used for Scientific Purposes. Hippocampi were dissected in ice-cold Gey's Balanced Salt Solution (Merck) supplemented with 28 mM glucose, 1 mM Kynureic acid, and 10 mM MgCl₂, and 350 μm hippocampal slices were cut using McIlwain tissue chopper. Slices were cultured on 0.4 μm Millicell membrane inserts (Merck) in Minimum Essential Medium (MP Biomedicals) supplemented with 25% Hank's Balanced Salt Solution (MP Biomedicals), 25% horse serum, 1% penicillin–streptomycin, 1 mM GlutaMax (all Thermo Fisher Scientific), and 28 mM glucose (Sigma-Aldrich). Medium was changed three times per week. After 4 DIV, cultures were transfected with plasmids using a biolistic method (Helios Gene Gun, Bio-

Rad). To obtain sparse astrocyte labelling we used 1 µm gold particles (Bio-Rad) and followed a standard protocol (*Benediktsson et al., 2005*) for preparation of gene gun bullets. Slices were shot at 160 PSI Helium pressure using modified gene gun barrel, in accord with accepted routines (*Woods and Zito, 2008*), where diffuser screen was replaced with stainless steel wire mesh (180 mesh per inch, 36% open area; Advent Research Materials Ltd.). Slices were used for experiments 4–10 days after transfection.

## Immunostaining in cultures

Cultures transfected with GLT1-SEP or GLT1ΔC-SEP were fixed using 37°C pre-warmed 4% paraformaldehyde in PBS for 10 min at 37°C, washed thrice in PBS, cell-permeabilised in 0.1% Triton X-100 for 10 min and blocked with 3% BSA, and 10% normal goat serum in PBS for 1 hr at RT. Afterwards, the cultured were incubated with the primary antibody (see below) in 1.5% BSA and 5% normal goat serum in PBS overnight at 4°C, washed thrice with PBS, incubated with the secondary antibody (see below) in 1.5% BSA and 5% normal goat serum in PBS for 2 hr, and washed thrice with PBS. The cultures were mounted using Fluoromount-G Mounting Medium (Thermo Fisher Scientific #00-4958-02) and stored at 4°C until imaging.

The primary antibodies used were: glial glutamate transporter GLT1 (guinea pig, polyclonal, synthetic peptide from the C-terminus of rat GLT1, Merck, AB1783, AB_90949, dilution 1:1,000) and GFP (chicken, polyclonal, GFP directly from *Aequorea Victoria*, Thermo Fisher Scientific, A10262, AB_2534023, dilution 1:1,000). Secondary antibodies: anti-chicken IgY (goat, Alexa Fluor 488, Thermo Fisher Scientific, A32931, dilution: 1:1,000) and anti-guinea pig IgG (goat, Alexa Fluor 647, Thermo Fisher Scientific, A-21450, dilution: 1:1,000). Images were recorded with Zeiss 780 confocal microscope using 40×/1.4 NA Oil Plan-Apochromat objective. The analyses were carried out in Fiji (NIH) using in-house written custom macros. The GLT1 fluorescence levels were measured in transfected cells versus non-transfected using the mask of GFP fluorescence for separation between the two groups. The mean fluorescence intensity was measured and compared in situ, for transfected and non-transfected, for each individual ROI image.

## Imaging and FRAP

Imaging was performed using an Olympus FV1000 system under Olympus XLPlan N25 × water immersion objective (NA 1.05). Imaging system was linked to two mode-locked, femtosecond-pulse Ti:Sapphire lasers (MaiTai from SpectraPhysics-Newport and Chameleon from Coherent), first one for imaging, was set at a wavelength of 910 nm and the other was for bleaching set on 690 nm, each of the lasers was connected to the microscope via an independent scan head. 690 nm for bleaching was selected based on the 2 P excitation spectrum for GFP (*Drobizhev et al., 2011*). The imaging laser power was kept below 4 mW under the objective at all times to minimise phototoxic damage, a power range validated by us previously in similar settings (*Jensen et al., 2019*). Bleaching laser power was kept around 10 mW. Dissociated mixed cultures were imaged in extracellular solution containing: 125 mM NaCl, 2.5 mM KCl, 30 mM glucose, 25 mM HEPES, 2 mM CaCl$_2$, and 1.3 mM MgSO$_4$; pH 7.4, at 32–34°C. In puffing experiments, pH 5.5 extracellular solution contained: 125 mM NaCl, 2.5 mM KCl, 30 mM glucose, 25 mM MES, 2 mM CaCl$_2$, and 1.3 mM MgSO$_4$, extracellular solution with 50 mM NH$_4$Cl contained: 50 mM NH$_4$Cl, 75 mM NaCl, 2.5 mM KCl, 30 mM glucose, 25 mM HEPES, 2 mM CaCl$_2$ and 1.3 mM MgSO$_4$, pH 7.4.

Organotypic cultures were imaged in artificial cerebrospinal fluid (aCSF) containing: 125 mM NaCl, 2.5 mM KCl, 2 mM CaCl$_2$, 1.3 mM MgSO$_4$, 26 mM NaHCO$_3$, 1.25 mM NaH$_2$PO$_4$, 20 mM D-glucose, 0.2 mM Trolox, bubbled with 95:5 O$_2$/CO$_2$ (pH 7.4) at 32–34°C.

FRAP experiments were performed using 22× zoom at 256 × 256 numerical resolution, resulting in a ~0.09 µm pixel size. Frame size was kept constant: 138 × 80 pixels, giving 148.32 ms per frame (unidirectional scanning, 4.0 µs pixel dwell time). The small (~1.6 µm wide) FRAP ROIs were selected in a quasi-random fashion. As in the hippocampal area CA1 the mean inter-synaptic distance is ~0.5 µm (*Rusakov and Kullmann, 1998*), each ROIs was equally likely to include one to three PSDs and the perisynaptic areas, thus providing a relatively homogenous sampling. The bleached ROI region was thus kept constant – a 18-pixels diameter circle (2.06 µm$^2$) scanned with second laser using tornado mode, resulting in fast bleaching time – 46 ms. In some experiments, drugs (1 mM glutamate or 100 µM ATP) were puffed for 250 ms just before the bleaching using Pneumatic PicoPump (World

Precision Instruments). Imaging, bleaching, and puffing were synchronised using Axon Digidata digitiser (Molecular Devices).

For whole cell FRAP (only in dissociated culture) 512 × 512 pixel frames were imaged every 1.644 s. Bleached region – 398-pixels diameter circle was scanned with second laser using tornado mode resulting in fast bleaching time – 2.00 s. In order to image and bleach as big astrocyte surface as possible, we used 2–5× zoom resulting in corresponding pixel size 0.497 µm to 0.198 µm. Pixel size was taken into account for data analysis and calculations.

In some FRAP experiments (see Results) we used the following drugs in the bath solution: TTX (1 µM, Tocris), MPEP (1 µM, Tocris), NBQX (10 µM, Tocris), LY 341495 (30 nM, Tocris), YM (300 nM, Tocris), Bicuculine (10 µM, Sigma-Aldrich), 4-AP (4-Aminopyridine, 500 µM, Sigma-Aldrich), FK-506 (1 µM, Sigma-Aldrich), and Staurosporine (100 nM, Cell Signaling Techn.).

## Super-resolution microscopy

We used the single-molecule localisation microscopy (SMLM) technique direct stochastic optical reconstruction microscopy (dSTORM) (*van de Linde et al., 2011*; *Endesfelder and Heilemann, 2015*) as described previously (*Heller et al., 2017*; *Heller and Rusakov, 2019*; *Heller et al., 2020*). Naïve dissociated hippocampal cultures and cultures expressing either GLT1-SEP or GLT1ΔC-SEP were fixed using 37°C pre-warmed 4% paraformaldehyde in PEM buffer (80 mM PIPES pH 6.8, 5 mM EGTA, 2 mM $MgCl_2$) (*Leyton-Puig et al., 2016*; *Pereira et al., 2019*) for 10 min at 37°C. Then, cells were washed thrice in PBS, incubated in 0.1% $NaBH_4$ in PBS for 7 min, washed thrice with PBS, and incubated in 10 mM $CuSO_4$ in 50 mM $NH_4Cl$, final pH = 5 for 10 min. Cells were washed thrice with water quickly and once with PBS. Cells were then permeabilised and blocked with PBS-S (0.2% saponin in PBS) supplemented with 3% BSA for 1 hr. Afterwards, cells were incubated with primary antibody (see below) in PBS-S overnight at 4°C, washed thrice with PBS-S, incubated with secondary antibody (see below) in PBS-S for 2 hr, washed twice with PBS-S and twice with PBS. Lastly, cells were post-fixed with 4% paraformaldehyde in PBS, washed thrice with PBS, and stored at 4°C until being prepared for imaging.

Primary antibodies used: post-synaptic protein PSD-95 (mouse, 6G6-1C9, recombinant rat PSD-95, Novus Biologicals, NB110-61643, AB_965165, dilution 1:500), glial glutamate transporter GLT1 (guinea pig, polyclonal, synthetic peptide from the C-terminus of rat GLT1, Merck, AB1783, AB_90949, dilution 1:1,000), and GFP (chicken, polyclonal, GFP directly from *Aequorea Victoria*, Thermo Fisher Scientific, A10262, AB_2534023, dilution 1:1,000).

Secondary antibodies used: anti-mouse IgG (donkey, CF568-conjugated, Biotium, 20105, AB_10557030, dilution 1:500), anti-chicken IgY (goat, Alexa647-conjugated, Thermo Fisher Scientific, A21449, AB_1500594, dilution: 1:1,000), and anti-guinea pig IgG (donkey, Alexa647-conjugated, Jackson ImmunoResearch Labs, 706-606-148, AB_2340477, dilution: 1:1,000).

Images were recorded with a Vutara 350 microscope (Bruker) in photo-switching buffer containing 100 mM cysteamine and oxygen scavengers (glucose oxidase and catalase) (*Metcalf et al., 2013*). Images were recorded with frame rate of 33 Hz (561 nm for CF568) or 66 Hz (640 nm for Alexa647). Total number of frames acquired per channel ranged from 3000 to 20,000. Data were analysed using the Vutara SRX software (version 6.02.05). Fiducial markers (100 nm TetraSpeck microspheres, T7279, Thermo Fisher Scientific) were used for drift correction.

## Cluster and nearest-neighbour analysis

In dSTORM maps, clusters of PSD95 were identified using DBScan, a well-established density-based clustering algorithm (*Ester et al., 1996*), with a minimum of 50 particles per cluster and a maximum particle distance of 100 nm; the latter parameters correspond to 250–300 nm wide PSD5 clusters which are consistent with the typical PSD size at common central synapses (*Chen et al., 2008*). The total 2D area for analysis for each GLT1 variant was ~360 µm², the cut-off distance for nearest-neighbour analyses was 500 nm, sampled within the 1 µm wide ROIs selected arbitrarily. Overall, the WT GLT1 samples contained 876,904 molecular positions and 22,887 nearest-neighbour distance measurements, GLT1-SEP samples contained 715,977 positions and 5021 distances, and GLT1ΔC-SEP samples contained 457,221 positions and 7193 distances.

The distribution of nearest-neighbour distances $D(r)$ between PSD95 clusters and GLT1 molecular species (and also among GLT1 molecular species) was calculated as the occurrence of distances $r$,

with a 5 nm or 10 nm binning step, and normalised to the overall number of registered events. To assess non-uniformity of the experimental distribution pattern $D(r)$ was compared to the theoretical $D(r)$ of a 2D Poisson point process (evenly random scatter) of the same surface density $\lambda$, in the form $D(r) = 1 - exp(-\lambda\pi r^2)$ (*Stoyan, 2006*).

## FRAP data analysis

Raw images were analysed using ImageJ. Mean fluorescence intensity was calculated for manually selected ROIs: Background – manually selected background ROI outside of transfected cell (FBKG), reference ROI which was manually outlined transfected cell in the imaged frame (FREF). For each frame mean fluorescence intensity of bleached ROI (FBL) was normalised according to the formula: F_NOR=(F_BL-F_BKG)/(F_REF-F_BKG). Normalised fluorescence value at the frame after the bleaching pulse (close to the background value) was subtracted from all values in data set. Finally, resulting fluorescence values were normalised to 40 frames before bleaching.

For the whole-cell bleaching experiments we performed similar analysis; however, for each cell we have measured mean fluorescence in manually selected three ROIs (FROI) defined as ~10 μm diameter circle placed outside of the cell soma. Additionally for each analysed data set we measured mean fluorescence in manually selected background ROI (FBKG) – outside of transfected cell and in reference ROI (FREF) which was manually outlined transfected cell outside of bleached region. We performed the same normalisation for each specific ROI as described above. The kinetic analyses of membrane turnover and FRAP traces were carried out as detailed in *Figure 1—figure supplements 1*, *2*. The FRAP time course $C_{mem}^f = R \cdot C_{in}(e^{-k_b t} - e^{-k_1 t})$ (notations in *Figure 1—figure supplements 1*, *2*) was fitted using the non-linear fitting routines ExpGroDec (exponent fitting) in Origin (OriginLab).

To evaluate lateral diffusivity from the spot-FRAP kinetics, we used the well-established Soumpasis method for circular ROIs (*Soumpasis, 1983*; *Kang et al., 2009*), in which the fluorescence time course is fitted with the equation $F(t) = C_{mob} \cdot exp\left(-\frac{2\tau_D}{t}\right) \cdot \left(\mathbf{I}_0\left(\frac{2\tau_D}{t}\right) + \mathbf{I}_1\left(\frac{2\tau_D}{t}\right)\right)$, where $C_{mob}$ is the mobile fraction, $\tau_D = \frac{w^2}{4D}$, $w$ is ROI radius, $D$ is diffusion coefficient, and $\mathbf{I}_0$ and $\mathbf{I}_1$ are modified Bessel functions of the first kind; this fitting has only two free parameters, $C_{mob}$ and $D$. The fitting was carried out using Soumpasis in the Origin software (OriginLab). The average diffusivity $D^*$ was therefore calculated as $D^* = C_{mob} \cdot D$.

Statistical inference was calculated using Origin's Hypothesis Testing from individual ROIs as statistical units (2–4 per cell): routine two-way ANOVA tests indicated no significant influence of the cell identity factor on the effects of experimental manipulations under study.

## Acknowledgements

This study was supported by the Wellcome Trust Principal Fellowship (212251_Z_18_Z), ERC Advanced Grant (323113),European Commission NEUROTWIN grant (857562), to DAR; and by National Science Centre Poland (2017/26/D/NZ3/01017), to PM.

## Additional information

### Funding

| Funder | Grant reference number | Author |
| --- | --- | --- |
| Wellcome Trust | 212251_Z_18_Z | Dmitri A Rusakov |
| European Research Council | 323113 | Dmitri A Rusakov |
| European Commission | 857562 | Dmitri A Rusakov |
| National Science Centre Poland | 2017/26/D/NZ3/01017 | Piotr Michaluk |

The funders had no role in study design, data collection and interpretation, or the decision to submit the work for publication.

## Author contributions
Piotr Michaluk, Conceptualization, Formal analysis, Investigation, Visualization, Methodology, Writing - review and editing; Janosch Peter Heller, Formal analysis, Investigation, Visualization, Methodology, Writing - review and editing; Dmitri A Rusakov, Conceptualization, Resources, Formal analysis, Supervision, Funding acquisition, Writing - original draft, Project administration, Writing - review and editing

## Author ORCIDs
Piotr Michaluk (iD) http://orcid.org/0000-0003-2306-3314
Janosch Peter Heller (iD) https://orcid.org/0000-0002-8825-3787
Dmitri A Rusakov (iD) https://orcid.org/0000-0001-9539-9947

## Ethics
Animal experimentation: Animal experiments were carried out in accordance with the national guideline and the European Communities Council Directive 0f November 1986, the European Directive 2010/63/EU on the Protection of Animals used for Scientific Purposes, and the United Kingdom Home Office (Scientific Procedures) Act of 1986, under UK Home Office Project licence PPL707524.

## Decision letter and Author response
Decision letter https://doi.org/10.7554/eLife.64714.sa1
Author response https://doi.org/10.7554/eLife.64714.sa2

# Additional files

## Supplementary files
• Transparent reporting form

## Data availability
All data generated or analysed during this study are included in the manuscript and supporting files. Source data files have been provided for each corresponding Figure.

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
