## [Decision Letter]

**Acceptance summary:**

This manuscript presents interesting and valuable information concerning the dynamics of the main glial glutamate transporter GLT1 in hippocampal astrocytes. By uptaking extracellular glutamate, GLT1 serves to sharpen the spatial confinement of glutamatergic synaptic communication between neurons. Characterizing the membrane surface dynamics of GLT1 and its spatial distribution with respect to the location of synapses, therefore, is important for a better understanding of the mechanisms underlying the fine-tuning of synaptic transmission, the dysfunction of which can lead to pathological conditions. The present study used GLT1 tagged with a pH-sensitive fluorescent reporter (GLT1-SEP) along with powerful optical approaches and analysis tools to access the surface lateral mobility of GLT1-SEP and the exchange between surface and intracellular pools, as well as the nanoscale localization of GLT1 with respect to glutamatergic synapses. The authors find that a large proportion of surface GLT1-SEP undergoes turnover with a surface lifetime of 22 s, whereas a smaller fraction (~25%) remains largely immobile, whose proportion is further decreased by elevated activity. They also show that the cytoplasmic domain of GLT1-SEP helps to attenuate the basal turnover of surface GLT1, facilitate its proximal localization to synapses, and is required for activity-dependent increase in the mobile fraction. These results underscore the importance of the GLT1 C-terminus in the membrane turnover as well as in the activity-dependent lateral diffusion of the transporter at the plasma membrane. The experiments and analysis are well conducted and provide good support to the present conclusions.

**Decision letter after peer review:**

Thank you for submitting your article "Rapid recycling of glutamate transporters on the astroglial surface" for consideration by *eLife*. Your article has been reviewed by 3 peer reviewers, one of whom is a member of our Board of Reviewing Editors, and the evaluation has been overseen by Olga Boudker as the Senior Editor. The following individual involved in review of your submission has agreed to reveal their identity: Alexei Verkhratsky (Reviewer #2).

The reviewers have discussed the reviews with one another and the Reviewing Editor has drafted this decision to help you prepare a revised submission.

Summary:

In this study, Michaluk et al. examined the membrane dynamics of the main glial glutamate transporter GLT1 in hippocampal astrocytes, which was previously shown to shape synaptic transmission through regulating extracellular levels of glutamate. Using GLT1 tagged on its surface with a pH-sensitive fluorescent marker, GLT1-SEP, the authors performed (1) fluorescence recovery after photobleaching (FRAP) experiments to assess the basal and activity-dependent dynamics of surface GLT1-SEP and (2) super-resolution dSTORM imaging to determine the relationship between GLT1 and PSD-95, an excitatory synapse marker. A large proportion of surface GLT1-SEP underwent turnover with a surface lifetime of 22 s, whereas a smaller fraction (~25%) remained largely immobile, which was decreased upon increased activity. Notably, the cytoplasmic domain of GLT1-SEP was shown to attenuate the basal turnover of surface GLT1 and to facilitate its proximal localization to synapses; moreover, GLT1 cytoplasmic domain was required for activity-dependent increase in the mobile fraction.

While previous studies using single molecule tracking have demonstrated a role for the lateral diffusion of GLT1 in controlling the recruitment of GLT1 near active synapses, the present study uses powerful optical approaches and analysis tools to access both the surface lateral mobility and the exchange between surface and intracellular pools of GLT1. Furthermore, characterization of the nanoscale organization of GLT1 relative to synapses and its dependence on the C-terminal domain of GLT1 is presented. Altogether, the results are interesting and valuable, and underscore the importance of the GLT1 C-terminus in the membrane turnover and in the activity-dependent lateral diffusion of the surface GLT1. Nevertheless, some of the conclusions are not strongly supported by the data shown, and a revision that fully addresses the following points is requested.

Essential revisions:

1. Whereas the authors are careful in the direct interpretation of the data, there is a tendency to overstate the conclusions. For example, the last sentence of the abstract require reconsideration as they did not measure extracellular glutamate, neither in health nor in disease. Similarly, such overstatements need to be toned down throughout the manuscript.

2. A comparison of the expression levels of GLT1-SEP versus GLT1-WT (at the minimum in HEK cells but ideally in astrocytes) should be shown. This information is needed for validating the use of GLT1-SEP as a tool to probe GLT1 dynamics, since the addition of the SEP tag could alter membrane turnover on its own, even if normalised transporter current seems normal. This could be done through immunolabelling using the anti-GLT1 antibody.

3. One should quantify the expression levels of GLT1-SEP relative to endogenous GLT1 in astrocytes, and also confirm that the exogenous expression of GLT1 variants used here does not affect the subcellular localization and dynamics of other astroglial cell membrane proteins.

4. Figure 3, dSTORM data: Please clarify which pool of GLT-1 is being targeted (surface vs. intracellular) and provide further details regarding the numbers of sampled ROIs and/or individual molecules / distances analysed.

5. Figure 5. The involvement of metabotropic glutamate receptors, calcium signalling, calcineurin and kinases were tested by bath applying the inhibitors, which would expect to affect neuronal activity also. Hence, the results as shown cannot be solely attributed to astrocyte signalling. Either the effects of directly interfering with astrocyte signalling should be tested or the conclusions need to be toned down.

---

## [Author Response]

Essential revisions:1. Whereas the authors are careful in the direct interpretation of the data, there is a tendency to overstate the conclusions. For example, the last sentence of the abstract require reconsideration as they did not measure extracellular glutamate, neither in health nor in disease. Similarly, such overstatements need to be toned down throughout the manuscript.

We appreciate this comment. We note, however, that the role of GLT1 in controlling extrasynaptic glutamate escape (not to confuse with ambient glutamate concentration), has long been established, and that we have been exploring this quantitatively for some time (e.g., Rusakov and Kullmann 1998 J Neurosci 18: 3158; Scimemi et al. 2004 J Neurosci 24: 4767; Sylantyev et al., 2008 Science 319:1845; Savtchenko et al. 2013 Nat Neurosci 16:10). In particular, our recent study (Henneberger et al. 2020 Neuron 108:919) uses a variety of methods to show, in a relatively direct fashion, how perisynaptic GLT1 control glutamate escape. The abstract wording has been adjusted, and we have also toned down other related claims.

2. A comparison of the expression levels of GLT1-SEP versus GLT1-WT (at the minimum in HEK cells but ideally in astrocytes) should be shown. This information is needed for validating the use of GLT1-SEP as a tool to probe GLT1 dynamics, since the addition of the SEP tag could alter membrane turnover on its own, even if normalised transporter current seems normal. This could be done through immunolabelling using the anti-GLT1 antibody.

We understand this general issue, and one reason for tagging an extracellular GLT1 domain was to minimise an interference with protein turnover, the latter being normally attributed to intracellular protein interactions. In this respect, we note that the FRAP of fluorophore-fused proteins has for decades served as a key method to grasp their membrane kinetics, and that fluorescent tagging has been a prominent feature of recent protein turnover studies (e.g., Alber et al. 2018 Mol Cell 71:1079; Rudolh et al. PNAS 2019 116: 25126). We note that, in the present context, earlier related studies (Murphy-Royal et al. 2015 Nat Neurosci 18:219, AlAwabdh et al. 2016 Glia 64:1252) used large quantum dots attached to GLT1 primary antibodies to monitor protein mobility, providing valuable and widely accepted observations.

To follow Reviewer's suggestions systematically, we have carried out additional experiments. Firstly, we compared immunostaining against GLT1 in HEK cells expressing GLT1 (n = 155) or GLT1-SEP (n = 188), together with mRFP1, and found that general GLT1 levels in GLT1-SEP-transfected cells were significantly lower (Figure 1—figure supplement 2A-B). Secondly, we compared expression of GLT1 variants in HEK cells using quantitative western blot (against α-tubulin; n = 6 samples per condition), showing that the expression of GLT1 was indeed twice as high as that of GLT1-SEP (or GLT1ΔC-SEP, Figure 1—figure supplement 2C-D). This was fully consistent with the ~50% lower transporter current in GLT1-SEP versus GLT1-expressing cells (Figure 1), suggesting that the key glutamate uptake properties of GLT1-SEP are indistinguishable from native.

Next, we stained astrocytes in mixed hippocampal culture expressing either GLT1-SEP or GLT1ΔC-SEP, with GFP in the green channel while using an GLT1 C-terminus antibody in the red channel. We were thus able to compare a total GLT1 expression in the astrocytes transfected with GLT1-SEP (n = 29), with GLT1 expression in the neighbouring, nontransfected cells (Figure 1—figure supplement 3A). In the transfected astrocytes, the all-variant GLT1 level was higher than in non-transfected cells (Figure 1—figure supplement 3B). In the cells transfected with GLT1ΔC-SEP (n = 27), we detected only endogenous GLT1 levels as the antibody does not recognise GLT1ΔC-SEP.

We also confirmed that endogenous GLT1 expression level was stable among transfected culture samples in non-transfected cells (Figure 1—figure supplement 3C). If anything, these data suggested that GLT1-SEP or GLT1ΔC-SEP were expressed without significant interruption, representing a significant proportion, or possibly the majority, of GLT1 expressed variants. In addition, our FRAP experiments revealed little difference in membrane vs. intracellular fraction between GLT1-SEP and GLT1ΔC-SEP (Figure 2B and Figure 2C).

We have added these data to the revision. While these observations help provide a general picture of GLT1-SEP and GLT1ΔC-SEP expression in relation to that of native GLT1, such data, or indeed any other similar data, cannot on their own reveal subtle influences of having the SEP tag on the GLT1 kinetics. Yet we would like to stress here, once again, that the majority of well-established direct experimental interventions, such as patch-clamp, genetically encoded sensors, opsins, or fluorescent tags, by definition introduce some perturbation to the system. This has not, however, diminished the value of direct observations obtained with such tools. The text has been appended with the corresponding explanations.

3. One should quantify the expression levels of GLT1-SEP relative to endogenous GLT1 in astrocytes.

This request echoes the previous one. To quantify this unbiasedly, one has to label GLT1SEP and WT GLT1 in the same cell using two antibodies that recognise specifically these two variants, which does not appear technically feasible. Even if this was achievable, it is not clear how this would help to understand better the kinetics of GLT1 or GLT1-SEP, which is our present objective. Nonetheless, our observations contain strong circumstantial evidence arguing for the physiological relevance of our approach. Firstly, transfected astroglia show a consistently dense, fairly homogenous membrane-surface expression of GLT1-SEP, as reported by its fluorescent signal (Figure 1D, Figure 2) or by GFP staining (Figure 1—figure supplement 3A). This expression pattern is similar to the native GLT1 expression in astrocytes, which has been widely reported. Secondly, our dSTORM data indicate that, again, both GLT1-SEP transfected and non-transfected cells showed a similar singlemolecule distribution pattern with respect to PSD clusters (Figure 3). In contrast, the expression pattern of GLT1ΔC-SEP, which does interfere with the protein turnover kinetics, was strikingly different (highly clustered; Figure 3).

And also confirm that the exogenous expression of GLT1 variants used here does not affect the subcellular localization and dynamics of other astroglial cell membrane proteins.

We are slightly puzzled by this request because astroglial membranes contain 1000s of different proteins, which would not appear feasible to explore and characterise to any significant degree. As noted above, the expression of GLT1-SEP was consistently dense and relatively homogenous across the entire astrocyte surface, similar to that of native GLT1. With no difference between such surface-filling expression patterns, it is difficult to conceive any difference in the physical effects of the protein in question on the distribution of other membrane proteins: in both cases, the cell membrane is overcrowded with the transporter proteins. To fit a promoter into the plasmid while possibly minimising interference with the regulatory mechanisms of expression, we used a short version of GFAP promoter, which has long been considered a relevant tool to explore protein expression in astrocytes (Lee et al. 2008 *Glia* 56: 481-493). We have added further explanations to the Discussion.

4. Figure 3, dSTORM data: Please clarify which pool of GLT-1 is being targeted (surface vs. intracellular).

The dSTORM protocols used permeabilization hence both surface and intracellular GLT1 variants were targeted. However, because the intracellular protein fraction is only 25-30%, the respective majority of dSTORM label reports the surface fraction. We have added the corresponding clarification.

And provide further details regarding the numbers of sampled ROIs and/or individual molecules / distances analysed.

The total 2D area for analysis for each GLT1 variant was ~360 μm^2^, the cut-off distance for nearest-neighbour analyses was 500 nm, sampled within the 1 μm wide ROIs selected arbitrarily (~36 per condition). Overall, the WT GLT1 and PSD labels represented 876903 molecular positions and 22887 nearest-neighbour distance measurements, GLT1-SEP samples contained 715977 positions and 5021 distances, GLT1ΔC-SEP samples contained 457221 positions and 7193 distances. These details have been added to the Materials and methods.

5. Figure 5. The involvement of metabotropic glutamate receptors, calcium signalling, calcineurin and kinases were tested by bath applying the inhibitors, which would expect to affect neuronal activity also. Hence, the results as shown cannot be solely attributed to astrocyte signalling. Either the effects of directly interfering with astrocyte signalling should be tested or the conclusions need to be toned down.

While we understand this concern, there must have been some misunderstanding. Our task was to test how the 4-AP-induced boost in neuronal activity affects GLT1 mobility in individual astroglia. We therefore compared GLT1 mobility before and after LP-4 application, under exactly the same pharmacological challenge. At the same time, the pharmacological manipulations per se had no detectable influence on the prominent boosting effect of 4-AP on network activity (as in Figure 5B). We therefore consider that, in the present experimental design, any concomitant effects of drug application on neurons were largely cancelled out, for comparison purposes. We have amended the text to clarify this further.